# Informed consent rates for neonatal randomized controlled trials in low- and lower middle-income versus high-income countries: A systematic review

**Jacquelyn K. Patterson**[1]*, **Stuti Pant**[2°], **Denise F. Jones**[1°], **Syed Taha**[2°], **Michael S. Jones**[3], **Melissa S. Bauserman**[1], **Paolo Montaldo**[2], **Carl L. Bose**[1], **Sudhin Thayyil**[2]

1 Department of Pediatrics, University of North Carolina School of Medicine, Chapel Hill, North Carolina, United States of America, 2 Centre for Perinatal Neuroscience, Imperial College London, London, United Kingdom, 3 Department of Agricultural and Resource Economics, North Carolina State University, Raleigh, North Carolina, United States of America

° These authors contributed equally to this work.
* Jackie_Patterson@med.unc.edu

**Data Availability Statement:** All relevant data are within the manuscript and its Supporting information files.

## Abstract

### Objective

Legal, ethical, and regulatory requirements of medical research uniformly call for informed consent. We aimed to characterize and compare consent rates for neonatal randomized controlled trials in low- and lower middle-income countries versus high-income countries, and to evaluate the influence of study characteristics on consent rates.

### Methods

In this systematic review, we searched MEDLINE, EMBASE and Cochrane for randomized controlled trials of neonatal interventions in low- and lower middle-income countries or high-income countries published 01/01/2013 to 01/04/2018. Our primary outcome was consent rate, the proportion of eligible participants who consented amongst those approached, extracted from the article or email with the author. Using a generalised linear model for fractional dependent variables, we analysed the odds of consenting in low- and lower middle-income countries versus high-income countries across control types and interventions.

### Findings

We screened 3523 articles, yielding 300 eligible randomized controlled trials with consent rates available for 135 low- and lower middle-income country trials and 65 high-income country trials. Median consent rates were higher for low- and lower middle-income countries (95.6%; interquartile range (IQR) 88.2–98.9) than high-income countries (82.7%; IQR 68.6–93.0; p<0.001). In adjusted regression analysis comparing low- and lower middle-income countries to high-income countries, the odds of consent for no placebo-drug/nutrition trials

**Funding:** The authors received no specific funding for this work.

**Competing interests:** The authors have declared that no competing interests exist.

was 3.67 (95% Confidence Interval (CI) 1.87–7.19; p = 0.0002) and 6.40 (95%CI 3.32–12.34; p<0.0001) for placebo-drug/nutrition trials.

## Conclusion

Neonatal randomized controlled trials in low- and lower middle-income countries report consistently higher consent rates compared to high-income country trials. Our study is limited by the overrepresentation of India among randomized controlled trials in low- and lower middle-income countries. This study raises serious concerns about the adequacy of protections for highly vulnerable populations recruited to clinical trials in low- and lower middle-income countries.

## Introduction

Legal, ethical and regulatory requirements of medical research uniformly call for informed consent [1–3]. Informed consent is defined as "authorization of an activity based on an understanding of what that activity entails and in the absence of control by others" [1, 4]. Informed consent is based on the principle of autonomy.

Recent events have highlighted legal and regulatory implications for researchers regarding informed consent [5, 6]. An unexpectedly high mortality rate was observed in one study arm of the Surfactant, Positive Pressure, and Oxygenation Randomized Trial (SUPPORT) in the United States, a trial comparing two different levels of oxygen targeting in premature babies. This led to prolonged litigation as the risk of increased mortality was not explicitly mentioned in the consent form [7]. Additionally, concerns have been raised about pharmaceutical-sponsored paediatric clinical trials enrolling in India without adequate informed consent, leading to serious harm and death of participants [5, 6, 8]. These concerns prompted the government of India to tighten regulatory approval of research and to mandate audio-visual recording of the informed consent process [9].

In the context of clinical trials, informed consent is the result of an interaction between the researcher and a potential, eligible participant. The informed consent process involves disclosure by the researcher, followed by comprehension, voluntary choice and authorization by the potential participant [1]. First, the researcher discloses complete and understandable information about the research and participant's rights, and invites questions [10]. This exchange results in the potential participant's true understanding of what is being asked. The potential participant then freely chooses to decline or authorize participation. The informed consent rate, the percentage of those enrolled among those approached, is one indicator of this informed consent process.

Extremely high consent rates for randomized clinical trials (RCTs) could reflect a large gap between the theory and practice of informed consent. Anecdotal experience conducting clinical trials in low- and lower middle-income countries (LMICs) suggests that consent rates are consistently high. In this systematic review, we compare informed consent rates for neonatal RCTs in LMICs versus high-income countries (HICs), and examine the influence of study characteristics on consent rates.

## Materials and methods

We conducted a systematic review of neonatal RCTs to evaluate consent rates in LMICs versus HICs.

## Search strategy and selection criteria

On April 1, 2018, we searched MEDLINE, EMBASE and Cochrane for neonatal RCTs of any intervention in LMICs or HICs published between January 1, 2013 and April 1, 2018. We used the following MEDLINE MeSH terms: 1) infant, 2) newborn OR infant. We also included the search terms randomized controlled trial(s) OR RCT(s). For the LMIC search, we paired descriptors related to economic status (e.g., low income, middle income, developing) with descriptors related to the population. We also searched for LMICs using individual country names per the 2017 World Bank criteria [11, 12]. For the HIC search, we paired descriptors related to economic status (e.g., high income, developed) with descriptors related to the population, and searched for HICs using individual country names (full search string included in S1 Table) [12]. In January 2021, we repeated the literature search using the same search strategy and retrieved a random sample of 20 neonatal RCTs (10 trials from LMICs and 10 trials from HICs) published between January 1, 2020 and January 1, 2021.

The search was conducted by a specialist librarian. Since the number of articles for HICs far exceeded those for LMICs, we used a random number generator to select a subset from the HIC group for screening. Two reviewers independently screened all articles for inclusion starting with the title and abstract, and proceeding to full-text; a third reviewer adjudicated discrepancies. We used Covidence systematic review software for screening (Veritas Health Innovation, Melbourne, Australia; www.covidence.org).

We included studies that were RCTs, implemented a newborn intervention and occurred in an LMIC or HIC. We defined an RCT as any trial that randomized participants, including individual and cluster randomization. We defined an intervention on the newborn as one initiated during the first 28 days after birth. We excluded studies from upper middle-income countries as well as studies that extended enrolment or allowed for initiation of the intervention beyond the newborn period. We did not include follow-up studies of primary trials, secondary analyses, studies published in languages other than English, conference abstracts, commentaries or study protocols.

## Data analysis

We extracted the following study characteristics: 1) country where the RCT took place, 2) control type (no placebo, placebo), 3) intervention (drug/nutrition, medical device, other), 4) funding (public, private, both, none, not stated), 5) timing of consent (antenatal, postnatal, both), 6) publication year, 7) method of randomization (individual, cluster, quasi), and 8) number enrolled, displayed as trial size (very large [n>1000], moderate to large [n = 100–1000], small [n<100]). We defined a drug/nutrition intervention as any substance introduced into the body by enteral, nasal, inhaled, subcutaneous, topical or intravenous routes; this included nutrition interventions such as macronutrients, micronutrients, vitamins and food products (breastmilk, colostrum, formula, fortification). We defined medical device interventions as interventions involving medical equipment such as respiratory equipment, methods for obtaining vascular access and suction devices (of note, we included commercially available, approved devices in this category). All interventional trials that could not be classified as drug/nutrition or medical device were classified as other (e.g., delayed cord clamping). We defined public funding as any source of funds supported by taxes such as the National Institutes of Health, the Department for International Development, and the World Health Organization.

We also extracted data on the consent process including format of consent (written, written or thumbprint, strictly verbal, written or verbal, not stated), method of obtaining consent (informed consent, deferred informed consent, informed or deferred consent, not stated), and video recording of consent (yes, no or not stated). To understand the consent rate, we collected

the number approached and consented as well as the number actually enrolled. To minimize publication bias with respect to consent rate reporting, we requested the consent rates from primary authors of articles that did not report it. To explore whether our findings remain relevant in newly published trials, we extracted and analyzed consent rates only from the random sample of trials obtained from our more recent search.

Our primary outcome was consent rate, defined as the percentage of participants who consented amongst those who were eligible and approached. If additional factors made a participant ineligible between screening and enrolment, we presumed they were not approached for consent.

We analysed the difference in study characteristics between LMIC and HIC trials using the Chi-squared test. In descriptive analysis, we compared distribution of consent rates between LMIC and HIC trials overall, as well as among the sub-samples of control and intervention types using a two-sample Wilcoxon rank-sum test. We also compared the difference in sub-sample consent rates within LMICs and within HICs by control type using a two-sample Wilcoxon rank-sum test and by intervention and funding using the Kruskal-Wallis Rank test. We analysed the odds of consenting in LMICs versus HICs by trial type using a generalized linear model for fractional dependent variables, which specifies a logistic link function, binomial family distribution, and heteroscedasticity-robust standard errors [13]. Given the limited literature on variables affecting informed consent, we included all study characteristics collected unless there was limited variation between the groups (e.g., method of randomization). We used the continuous variable of number enrolled as our marker of trial size in the model. Our final model adjusted for funding, timing of consent, publication year and number enrolled (as a natural log). In supplementary material, we analysed the difference in study characteristics between trials reporting a consent rate and those that did not. We also analysed the odds of reporting a consent rate in LMICs versus HICs by trial type using logistic regression with the adjusted model including the study characteristics detailed above. Finally, we conducted a post-hoc analysis of the consent rate for HIC trials by region using the Kruskal-Wallis Rank test. We used STATA version 16 (College Station, TX) for all analyses.

There was no funding source for this study. The final protocol can be obtained by emailing the primary author. The corresponding author had full access to all the data in the study and had final responsibility for the decision to submit for publication. All authors approved the final version of the manuscript submitted for publication.

## Results

Our initial search identified 9270 publications, 7382 from HICs and 1888 from LMICs. Of the 7382 HIC publications, we selected 1763 for screening using a random number generator. A total of 1763 LMIC publications and 1760 HIC publications were screened for inclusion in this study (Fig 1). Of these, 193 LMIC trials and 107 HIC trials met our inclusion criteria. Consent rates were reported in 200 trials; 135 (69.9%) LMIC trials and 65 (60.7%) HIC trials (p = 0.106; two-sided t-test). The characteristics of the trials reporting consent rates were not significantly different compared to those not reporting consent rates, with the exception of more private funding among trials not reporting a consent rate (p = 0.028; Chi-square test; S2 Table).

The following results focus on the sample of studies with a consent rate; details of each included article are in the (S3 and S4 Tables). Control type, intervention, timing of consent and method of randomization were not statistically significantly different between LMIC and HIC trials. However, trials were significantly larger in LMICs versus HICs (p = 0.022; Chi-square test). There was also a significant difference in funding and publication year between LMIC and HIC trials, with LMIC trials often being unfunded and more recent (Table 1).

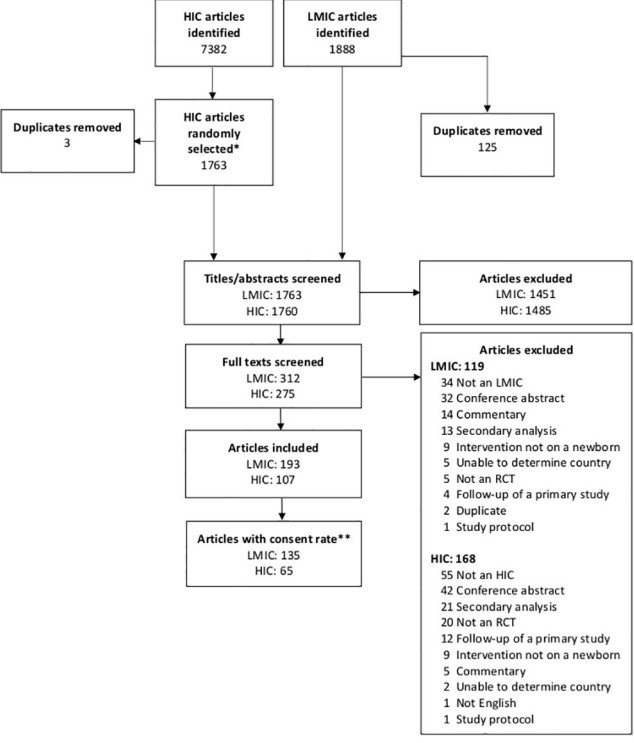

**Fig 1. Summary of search process.** Systematic search summary displayed as Consolidated Standards of Reporting Trials diagram.

India was the predominant location of trials in the LMIC group, and consent rates varied by country (Fig 2; S5 Table). Overall, consent rates in LMIC trials were higher than those in HIC trials with a median percent consented in LMICs of 95.6 (IQR 88.2–98.9) compared to 82.7 (IQR 68.6–93.0) in HICs (p<0.0001; two-sample Wilcoxon rank sum test; Table 2). LMIC trial consent rates were left-skewed towards high rates (Fig 3, Part A) and consistently higher than HIC rates at every quantile of the distribution (Fig 3, Part B). Furthermore, LMIC trials had consistently higher consent rates than HIC trials when comparing trials by control type or intervention (Table 2). Although consent rates within LMIC trials did not differ according to control type, intervention or funding, consent rates within HIC trials were significantly different based on both control type and funding (Table 3; Fig 4).

In multivariate regression modelling, odds of consent were about 3.6 times higher in LMICs versus HICs for drug/nutrition trials without a placebo and increased to 6.5 times higher when a placebo was used (Table 4). Statistically significant differences were not found in medical device or 'other' trials (all of which had no placebo designs). These relationships across income classification and trial type persisted in the adjusted model. In unadjusted and adjusted logistic regression models of all studies in this review, there was no systematic difference in the odds of reporting a consent rate based on income classification or trial type (S6 Table).

The majority of trials in both LMICs and HICs used written, informed consent (Table 5). One third of trials in this review did not state the format of consent. In light of the variation in HIC consent rates noted in this review, we examined HIC consent rates by region and found no regional differences in post-hoc analysis (p = 0.5087; Kruskal-Wallis Test; S1 Fig).

**Table 1. Study characteristics of neonatal randomized controlled trials by LMIC versus HIC.**

| Study characteristics | LMIC trials (N = 135) n (%) | HIC trials (N = 65) n (%) | p-value[a] |
|---|---|---|---|
| **Control type** | | | |
| No placebo | 110 (81.5) | 49 (75.4) | 0.317 |
| Placebo | 25 (18.5) | 16 (24.6) | |
| **Intervention** | | | |
| Drug/nutrition | 80 (59.3) | 36 (55.4) | |
| Medical device | 27 (20.0) | 14 (21.5) | 0.871 |
| Other | 28 (20.7) | 15 (23.1) | |
| **Funding** | | | |
| Public | 21 (15.6) | 13 (20.0) | |
| Private | 5 (3.7) | 16 (24.6) | |
| Both | 9 (6.7) | 9 (13.9) | <0.0001 |
| None | 49 (36.3) | 4 (6.2) | |
| Not stated | 51 (37.8) | 23 (35.4) | |
| **Timing of consent** | | | |
| Antenatal | 21 (15.6) | 4 (6.2) | |
| Postnatal | 108 (80.0) | 57 (87.7) | 0.159 |
| Both | 6 (4.4) | 4 (6.2) | |
| **Publication year** | | | |
| 2013 | 19 (14.1) | 20 (30.8) | |
| 2014 | 12 (8.9) | 14 (21.5) | |
| 2015 | 36 (26.7) | 13 (20.0) | |
| 2016 | 30 (22.2) | 10 (15.4) | 0.002 |
| 2017 | 28 (20.7) | 5 (7.7) | |
| 2018 | 10 (7.4) | 3 (4.6) | |
| **Method of randomization** | | | |
| Individual | 131 (97.0) | 65 (100.0) | |
| Cluster | 3 (2.2) | 0 (0.0) | 0.374 |
| Quasi | 1 (0.7) | 0 (0.0) | |
| **Trial size** | | | |
| Very large (n >1000) | 6 (4.4) | 0 | |
| Moderate to large (n = 100–1000) | 78 (57.8) | 27 (41.5) | 0.010 |
| Small (n <100) | 51 (37.8) | 38 (58.5) | |

[a]Chi-squared test used for all p-values.

## Discussion

In this systematic review of recently published neonatal RCTs, we found significantly higher rates of informed consent in LMICs compared to HICs. Notably, consent rates in LMICs were greater than 90% in two thirds of trials, as compared to only one quarter of trials in HICs. Although study characteristics such as funding and control type affected consent rates in HICs, these study characteristics had no impact on consent rates in LMICs. In adjusted regression modelling, potential participants had six times the odds of consenting to placebo-drug/nutrition studies in LMICs than in HICs.

This is the first systematic review comparing informed consent rates across a cohort of clinical trials in LMICs versus HICs. High rates of informed consent in LMICs have been

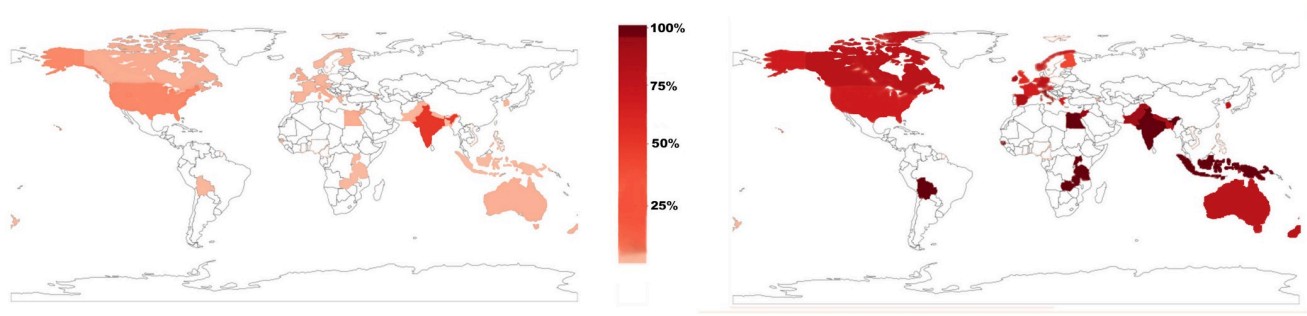

**Fig 2. Geographic distribution of included trials.** On the left, geographic distribution of trials in our study that reported a consent rate depicted as percent of trials. On the right, geographic distribution of consent rates depicted as median consent rate per country.

previously alluded to in the literature, and are commonly identified by researchers in LMICs [14]. For example, only 58% of researchers with trials in LMICs agreed that some potential participants declined to enrol after learning about the trial [15]. In contrast, it is not surprising that placebo-drug/nutrition studies had a median consent rate of 70% in HICs, the lowest consent rate among all trial types. This consent rate is consistent with previously reported consent rates of 48 to 79% in neonatal drug trials in HICs [16–19]. Motivators that decrease the likelihood of consent in paediatric HIC trials include randomization procedures, the use of placebos, high perceived risk, and anti-experimentation views [17–24]. In contrast, motivators that increase the likelihood of consent in such trials include obtaining access to new drugs or perceived superior treatment plans and shorter trials.

These data raise serious concerns about the adequacy of protections for highly vulnerable populations recruited for clinical trials in LMICs. Rates consistently in excess of 90% in trials that pose some risk suggest that there may be a flaw in the informed consent process. The trials in this review reported minimal data on the consent process itself, limiting our ability to evaluate whether these high consent rates are the result of inadequacies in the consent process. However, a number of previously identified factors in LMICs could adversely impact disclosure, comprehension, voluntary choice and authorization in the informed consent process.

**Table 2. Consent rates in LMIC versus HIC neonatal randomized controlled trials.**

| | LMIC trials, n = 135 | | HIC trials, n = 65 | | p-value |
|---|---|---|---|---|---|
| **Overall median % consented (IQR)** | 95.6 (88.2–98.9) | | 82.7 (68.6–93.0) | | <0.0001 |
| | **N** | **Median % consented (IQR)** | **N** | **Median % consented (IQR)** | **p-value** |
| **By control type** | | | | | |
| No placebo | 110 | 95.5 (88.2–99.2) | 49 | 85.9 (70.5–95.2) | <0.0001 |
| Placebo | 25 | 95.6 (89.6–97.4) | 16 | 70.0 (49.5–81.3) | <0.0001 |
| **By intervention** | | | | | |
| Drug/nutrition | 80 | 95.5 (90.0–98.9) | 36 | 79.3 (62.9–93.4) | <0.0001 |
| Medical device | 27 | 96.0 (83.1–98.6) | 14 | 86.3 (81.1–95.7) | 0.0745 |
| Other | 28 | 95.4 (80.6–99.1) | 15 | 83.3 (69.5–90.9) | 0.0433 |

Note: Two-Sample Wilcoxon rank-sum test used for all p-values. In our random sample of neonatal RCTs published between January 2020 and January 2021, the median percent consented in LMICs was 95.9 (IQR 94.4–96.1) whereas the median percent consented in HICs was 77.0 (IQR 66.5–87.5).

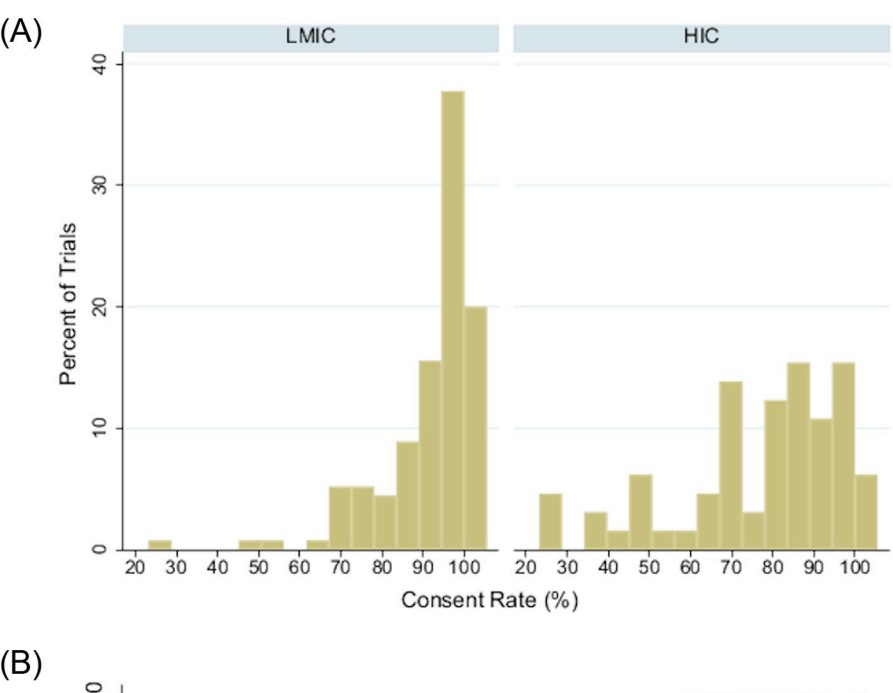

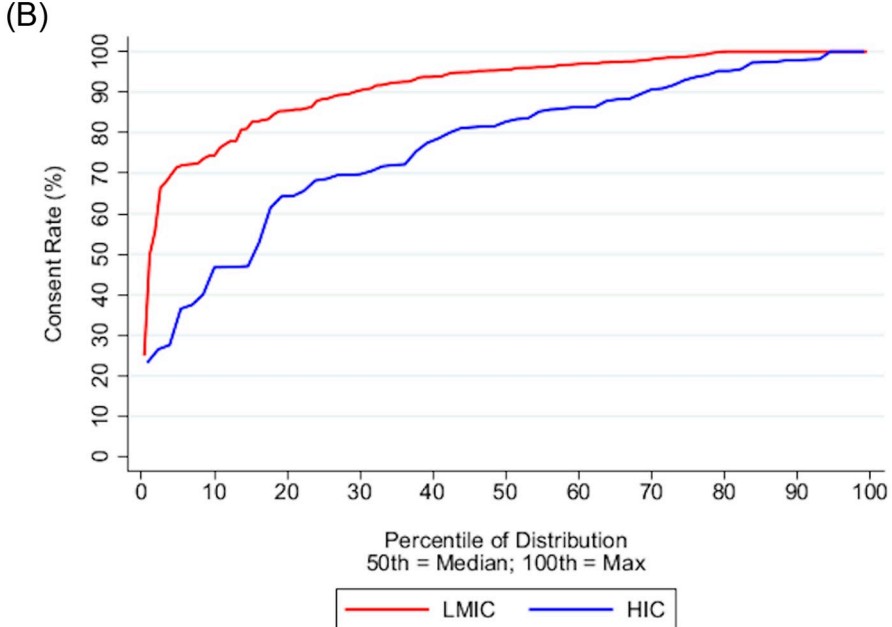

**Fig 3. Distribution of LMIC versus HIC trials by consent rate.** Part A: The histogram depicts the percent of LMIC versus HIC trials per consent rate increments of five percent. LMIC trials are left-skewed towards high consent rates. Part B: The cumulative density function depicts the percent of trials that have a consent rate equal to or less than y. Across the entire distribution, LMIC consent rates were consistently higher than HIC consent rates (p<0.0001; Two-Sample Wilcoxon Rank-Sum Test). 5% of LMIC trials had a consent rate less than 70% compared to one third of HIC trials.

## Disclosure

Institutional ethical review boards and research governance frameworks play an important role in ensuring that trials have well-defined and documented procedures for obtaining informed consent. This infrastructure is well-established in HICs, ensuring that research participants are informed and adequately protected while being recruited for and participating in

**Table 3. Effect of study characteristics on consent rates within LMIC and HIC trials.**

| Study characteristics | LMIC trials | | | HIC trials | | |
|---|---|---|---|---|---|---|
| | N | Median % consented (IQR) | p-value | N | Median % consented (IQR) | p-value |
| Control type[a] | | | | | | |
| No placebo | 110 | 95.5 (88.2–99.2) | 0.8217 | 49 | 85.9 (70.5–95.2) | 0.0110 |
| Placebo | 25 | 95.6 (89.6–97.4) | | 16 | 70.0 (49.5–81.3) | |
| Intervention[b] | | | | | | |
| Drug/nutrition | 80 | 95.5 (90.0–98.9) | 0.6574 | 36 | 79.3 (62.9–93.4) | 0.4754 |
| Medical device | 27 | 96.0 (83.1–98.6) | | 14 | 86.3 (81.1–95.7) | |
| Other | 28 | 95.4 (80.6–99.1) | | 15 | 83.3 (69.5–90.9) | |
| Funding[b] | | | | | | |
| Public | 21 | 92.0 (89.6–96.0) | 0.7720 | 13 | 70.5 (64.3–78.6) | 0.0003 |
| Private | 5 | 97.0 (95.2–97.4) | | 16 | 89.5 (70.6–96.5) | |
| Both | 9 | 97.1 (96.1–99.2) | | 9 | 47.1 (36.5–68.6) | |
| None | 49 | 96.7 (85.3–100.0) | | 4 | 94.5 (83.0–96.5) | |
| Not stated | 51 | 95.2 (88.2–98.5) | | 23 | 86.3 (81.3–93.0) | |

[a]p-values from Two-Sample Wilcoxon rank-sum test.
[b]p-values from Kruskal-Wallis rank test.

clinical trials. In contrast, the infrastructure in LMICs is often ill-equipped and poorly managed by overworked clinical staff or volunteers without specialist expertise [10]. Thus, it is possible that poor research governance in LMICs frequently leads to inadequate disclosure of the details of a study, a problem identified by at least 35% of researchers with trials in LMICs [15].

## Comprehension

Even if disclosure is adequate, individuals in LMICs may be at greater risk for poor comprehension. For example, participants may erroneously believe that they are consenting to receive treatment instead of to participate in research, a phenomenon referred to as therapeutic

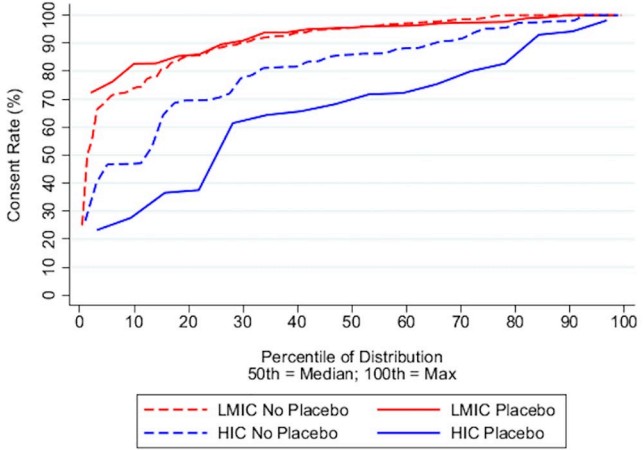

**Fig 4. Distribution of LMIC vs HIC trial consent rates by control type (no placebo versus placebo).** The cumulative density function depicts the percent of trials that have a consent rate equal to or less than each point on the y-axis. Across the entire distribution, HIC placebo trials had lower consent rates than HIC no placebo trials (p = 0.0111; Two-Sample Wilcoxon Rank Sum Test). There was little difference between the distribution of placebo and no placebo consent rates in LMIC trials (p = 0.8217; Two-Sample Wilcoxon Rank Sum Test).

**Table 4. Odds of consenting by trial type in LMIC versus HIC trials.**

| Control-intervention trial type | Unadjusted OR (95% CI) | p-value | Adjusted OR (95% CI) | p-value |
|---|---|---|---|---|
| Placebo–drug/nutrition | 6.54 (3.40–12.57) | <0.0001 | 6.40 (3.32–12.34) | <0.0001 |
| No placebo–drug/nutrition | 3.64 (1.89–7.01) | <0.0001 | 3.67 (1.87–7.19) | 0.0002 |
| No placebo–medical device | 1.67 (0.78–3.55) | 0.1857 | 1.77 (0.74–4.25) | 0.2030 |
| No placebo–other | 1.97 (0.92–4.23) | 0.0802 | 1.95 (0.95–4.00) | 0.0697 |

Note: N = 200 observations for both models. Unadjusted model with a Wald Chi–Square of 64.01 (p<0.0001) and Pseudo R-square of 0.0625. Adjusted model derived from fractional logistic regression adjusting for funding, timing of consent, publication year and log of number enrolled with a Wald Chi-Square of 106.68 (p<0.0001) and Pseudo R-square of 0.0820.

misconception [25–28]. Lower literacy levels and language barriers may also reduce comprehension in LMICs. Furthermore, poor comprehension may be more likely for the subset of clinical trials reviewed in this study, as prior work has demonstrated that comprehension of both randomization procedures and placebo controlled designs is generally lower than other aspects of trial comprehension [29–31]. However, this challenge in understanding complex trial designs is not unique to LMICs [29].

## Voluntary choice

Voluntary participation may be a challenging concept for potential participants in LMICs to understand. For example, evaluation of the consent process for an HIV study in a South African antenatal clinic revealed that 84% of participants thought their participation was compulsory [32]. Even when potential participants understand that involvement in research is voluntary, they may feel pressured to participate due to their child's illness [30]. This pressure may be disproportionately felt in LMICs where access to quality medical care is poor. Furthermore, even with standard of care or placebo control arms, participants may perceive additional incentives such as free health care, increased surveillance and reimbursement of travel costs as

**Table 5. Consent process among all included trials in LMICs versus HICs.**

| | LMIC trials (N = 193) n (%) | HIC trials (N = 107) n (%) |
|---|---|---|
| **Format** | | |
| Written | 117 (60.6) | 73 (68.2) |
| Written or thumbprint | 1 (0.5) | 0 |
| Strictly verbal | 2 (1.0) | 0 |
| Written or verbal | 2 (1.0) | 0 |
| Not stated | 71 (36.8) | 33 (30.8) |
| **Method** | | |
| Informed consent | 174 (90.2) | 96 (89.7) |
| Deferred informed consent | 0 | 0 |
| Informed or deferred consent | 2 (1.5) | 3 (2.8) |
| Not stated | 16 (8.3) | 8 (7.5) |
| **Video Recording** | | |
| Yes | 1 (0.5) | 1 (0.9) |
| No or not stated | 192 (99.5) | 106 (99.1) |

benefits to their participation. These incentives may hold greater importance as positive motivators for consent in LMICs.

## Authorization

Global guidelines for informed consent emphasize the importance of individual authorization for participation in research. This recommendation is founded upon the principle of autonomy which is a normative underpinning of the developed world. In contrast, communal consciousness in LMICs may raise the importance of involving community leaders and families in decision-making, and ultimately influence consent at the individual level [33]. In some cases, this can lead to a decision to participate before the informed consent process is even begun [34, 35]. Furthermore, vulnerable populations within LMICs with low socio-economic status and limited access to health care may pre-emptively decide to participate based on information elicited from the community [34]. Pre-emptive decisions to participate in advance of the informed consent process may alter the utility of this process.

Our study has a number of important limitations. The majority of LMIC studies were from India, thus may not be broadly representative of consent rates in other LMICs. We did not search trial registries for unpublished studies. We cannot comment on other factors beyond trial characteristics that have been associated with informed consent rates such as parental characteristics, physicians approaching participants for consent, or incentives to participation [36]. Finally, the LMIC trials included in this systematic review were larger than the HIC trials, raising the possibility that the LMIC trials disproportionately represented later phase trials with a more favourable benefit to risk ratio. However, this is unlikely to account for the difference in consent rates as our adjusted model controlled for study size; furthermore, study phase has not been found to impact consent rates [16].

Consent rates are an important indicator of the informed consent process, yet their utility in assessing the quality of the consent process is limited without additional information. Only 67% of studies that met inclusion criteria for this review reported adequate information to be able to calculate consent rates. Few additional details about the consent process were included in the methods of these papers, with one third of studies failing to report basic details such as the format of their consent. Although audio-visual recording of the consent process is a legal requirement in India since 2009, only one trial from India explicitly reported audio-visual consent. Global guidelines for reporting on clinical trials should include recommendations for standard reporting of both consent rates and relevant details of the consent process.

Our data suggest that the informed consent process is flawed in LMICs. Multiple international guidelines exist to govern informed consent processes globally [37–41]. Despite these guidelines, the informed consent process in LMICs may be fraught with inadequate disclosure by researchers, as well as poor comprehension and pressured authorization by participants. We join others in calling for an improvement in governance as well as culturally appropriate adaptations to the informed consent process in LMICs [10, 33]. The uniformly high consent rates in LMICs demonstrated in this review suggest that the need for such change is urgent. In fact, the ethical conduct of research in LMICs depends upon it.

## Supporting information

**S1 Fig. Trial consent rates by HIC regions.** There is no significant difference between median consent rates by region in HIC (p = 0.5087; Kruskal-Wallis test, excludes 'multiple').
(TIF)

**S1 Table. Search strategy for low- and lower middle-income and high-income country articles.**
(DOCX)

**S2 Table. Study characteristics among trials reporting a consent rate versus those that did not report a consent rate.**
(DOCX)

**S3 Table. Low- and lower middle-income country studies.**
(DOCX)

**S4 Table. High-income country studies.**
(DOCX)

**S5 Table. Consent rates by country.**
(DOCX)

**S6 Table. Odds of obtaining a consent rate (in article or by email response from study author) in LMIC versus HIC trials.**
(DOCX)

**S1 File.**
(DTA)

**S1 Checklist.**
(DOC)

## Acknowledgments

We would like to acknowledge Sarah Wright, the specialist librarian who developed and executed the search strategy.

## Author Contributions

**Conceptualization:** Jacquelyn K. Patterson, Denise F. Jones, Syed Taha, Melissa S. Bauserman, Carl L. Bose, Sudhin Thayyil.

**Data curation:** Jacquelyn K. Patterson, Denise F. Jones, Syed Taha.

**Formal analysis:** Stuti Pant, Michael S. Jones, Melissa S. Bauserman, Paolo Montaldo, Carl L. Bose, Sudhin Thayyil.

**Project administration:** Jacquelyn K. Patterson.

**Supervision:** Jacquelyn K. Patterson, Sudhin Thayyil.

**Writing – original draft:** Jacquelyn K. Patterson, Stuti Pant.

**Writing – review & editing:** Jacquelyn K. Patterson, Denise F. Jones, Syed Taha, Michael S. Jones, Melissa S. Bauserman, Paolo Montaldo, Carl L. Bose, Sudhin Thayyil.

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
