## [Decision Letter · Decision Letter 0]

12 Jan 2021

PONE-D-20-21677

Informed consent rates for neonatal randomized controlled trials in low- and lower middle-income versus high-income countries: A systematic review

PLOS ONE

Dear Dr. Patterson,

Thank you for submitting your manuscript to PLOS ONE. After careful consideration, we feel that it has merit but does not fully meet PLOS ONE’s publication criteria as it currently stands. Therefore, we invite you to submit a revised version of the manuscript that addresses the points raised during the review process.

A rebuttal letter that responds to each point raised by the  reviewer. You should upload this letter as a separate file labeled 'Response to Reviewers'.A marked-up copy of your manuscript that highlights changes made to the original version. You should upload this as a separate file labeled 'Revised Manuscript with Track Changes'.An unmarked version of your revised paper without tracked changes. You should upload this as a separate file labeled 'Manuscript'.

We look forward to receiving your revised manuscript.

Kind regards,

Ghislaine JMW van Thiel

Academic Editor

PLOS ONE

2. Please consider discussing the quality of each of the studies included in this review and their biases. In addition, please update your search to allow the inclusion of studies published in the past 12 months.

4. We note that Figure 2 in your submission contain map images which may be copyrighted. All PLOS content is published under the Creative Commons Attribution License (CC BY 4.0), which means that the manuscript, images, and Supporting Information files will be freely available online, and any third party is permitted to access, download, copy, distribute, and use these materials in any way, even commercially, with proper attribution. For these reasons, we cannot publish previously copyrighted maps or satellite images created using proprietary data, such as Google software (Google Maps, Street View, and Earth). For more information, see our copyright guidelines: http://journals.plos.org/plosone/s/licenses-and-copyright.

4.1.    You may seek permission from the original copyright holder of Figure 2 to publish the content specifically under the CC BY 4.0 license. 

4.2.    If you are unable to obtain permission from the original copyright holder to publish these figures under the CC BY 4.0 license or if the copyright holder’s requirements are incompatible with the CC BY 4.0 license, please either i) remove the figure or ii) supply a replacement figure that complies with the CC BY 4.0 license. Please check copyright information on all replacement figures and update the figure caption with source information. If applicable, please specify in the figure caption text when a figure is similar but not identical to the original image and is therefore for illustrative purposes only.

Reviewers' comments:

**Comments to the Author**

1. Is the manuscript technically sound, and do the data support the conclusions?

Reviewer #1: Yes

2. Has the statistical analysis been performed appropriately and rigorously? 

Reviewer #1: Yes

3. Have the authors made all data underlying the findings in their manuscript fully available?

Reviewer #1: Yes

4. Is the manuscript presented in an intelligible fashion and written in standard English?

Reviewer #1: Yes

5. Review Comments to the Author

Reviewer #1: Thank you for the opportunity to review this interesting manuscript reporting the differences in consent rates between LMIC and HIC for trials conducted with neonatal populations. This review is novel and important to inform research decisions. The manuscript is well written with comprehensive tables and figures supporting their analyses and conclusions. There are a few questions and concerns.

1. Please justify why middle-to-high income countries were excluded.

2. Was there a difference in rates of consent if mother vs father of the infant consented?

3. Similar to HIC, please provide acronym for LMIC early in the manuscript and use consistently.

4. Table 1. I struggled to find which study characteristics were significantly different. It would help the reader if a column with p values was added.

5. Except for a statement on line 299, the authors are largely silent on monetary and other incentives to participate in each study. Incentives may be critical to decisions to participate in trials, particularly in LMIC. If data were available for even a subset of included articles, consent by incentives this should be included as sub-analysis. If not, please address as a limitation of the review.

6. There may be a missing reference on line 285.

6. PLOS authors have the option to publish the peer review history of their article (what does this mean?). If published, this will include your full peer review and any attached files.

Reviewer #1: **Yes: **Karen M. Benzies

---

## [Author Response · Author response to Decision Letter 0]

8 Feb 2021

February 5, 2021

Ghislaine JMW van Thiel

Academic Editor

PLOS ONE

Dear Dr. van Thiel,

Thank you for your consideration of our manuscript titled Informed consent rates for neonatal randomized controlled trials in low- and lower middle-income versus high-income countries: A systematic review. We thank the editor and the reviewers for their helpful comments, and respectfully submit our revised manuscript. Below is a point-by-point response to the comments received, with our responses in blue italics.

Sincerely,

Jacquelyn Patterson, MD, MPH

Assistant Professor of Pediatrics

Journal requirements

We have formatted our manuscript according to the PLOS One style requirements, including those for file naming.

2. Please consider discussing the quality of each of the studies included in this review and their biases. In addition, please update your search to allow the inclusion of studies published in the past 12 months.

We did not assess the quality of the studies included since the quality of a study does not impact the validity of the informed consent rate reported. As such, traditional bias assessments were not relevant for this review of informed consent rates. 

We agree that it is important for systematic reviews and meta-analyses to be contemporary, ideally published within one year of the search date. This is to ensure there are no newly published trials which would influence the results of the meta-analysis, such as when reviewing the efficacy of a drug or prognostic accuracy of a test. 

In the case of this systematic review, we focused on all trials published within a particular time period, requiring us to screen approximately 3,500 trials and to extract data from approximately 200 full text articles. As such, this systematic review took us approximately 2 years to complete. Given the additional 6 months our publication has been under review at PLOS One, our search is now approximately 2.5 years old. Updating our search to include the past 2.5 years of trials will take at least one year to perform. 

To address the editor’s concern and to demonstrate that our review is sufficiently contemporary, we repeated the search and extracted consent rates in a random sample of 20 trials published in 2020. Our results demonstrate that consent rates have not changed since the end date of our previous search, and that the conclusions of our systematic review remain valid. We have included this data in both the methods and results section of the revised manuscript. 

In lines 91-93 and 130-132 of the methods, the revised manuscript includes the following new text: 

“In January 2021, we repeated the literature search using the same search strategy and retrieved a random sample of 20 neonatal RCTs (10 trials from LMICs and 10 trials from HICs) published between January 1, 2020 and January 1, 2021. To explore whether our findings remain relevant in newly published trials, we extracted and analyzed consent rates only from the random sample of trials obtained from our more recent search.”

Additionally, in table 2 under ‘Note’ in the results (lines 214-216), the revised manuscript includes the following new text: 

“In our random sample of neonatal RCTs published between January 2020 and January 2021, the median percent consented in LMICs was 95.9 (IQR 94.4-96.1) whereas the median percent consented in HICs was 77.0 (IQR 66.5-87.5).”

We would like to change our Data Availability statement so that it indicates that all data is provided as part of this publication. In this revision, we have included a csv file of our raw data as Supporting Information.

4. We note that Figure 2 in your submission contain map images which may be copyrighted. All PLOS content is published under the Creative Commons Attribution License (CC BY 4.0), which means that the manuscript, images, and Supporting Information files will be freely available online, and any third party is permitted to access, download, copy, distribute, and use these materials in any way, even commercially, with proper attribution. For these reasons, we cannot publish previously copyrighted maps or satellite images created using proprietary data, such as Google software (Google Maps, Street View, and Earth). For more information, see our copyright guidelines: http://journals.plos.org/plosone/s/licenses-and-copyright.

The map in Figure 2 of our manuscript was produced using R version 4.0 and the packages rnaturalearth and ggplot2. As such, copyrighting does not apply.

We have updated our manuscript accordingly.

At the time of acceptance to PLOS One, we will upload our protocol to this repository.

We have uploaded all figure files to PACE.

Comments to the Author

Thank you for the opportunity to review this interesting manuscript reporting the differences in consent rates between LMIC and HIC for trials conducted with neonatal populations. This review is novel and important to inform research decisions. The manuscript is well written with comprehensive tables and figures supporting their analyses and conclusions. There are a few questions and concerns.

1. Please justify why middle-to-high income countries were excluded.

Our hypothesis that prompted this systematic review was that consent rates in randomized controlled trials conducted in LMICs are much higher than in HICs. Unlike in HICs, LMICs often lack adequate research governance systems to protect trial participants. The situation in upper middle-income countries (UMICs) is much more variable, and the proportion of RCTs conducted in these settings was relatively small. As such, we elected to exclude UMICs from this review. 

2. Was there a difference in rates of consent if mother vs father of the infant consented?

Details regarding the consent process, including whether the mother or father provided consent, were limited in the articles included. As such, the available data does not allow us to answer this question.

3. Similar to HIC, please provide acronym for LMIC early in the manuscript and use consistently.

We provide an acronym for LMIC at its first use in the introduction, and have double checked that it is used consistently throughout the manuscript.

4. Table 1. I struggled to find which study characteristics were significantly different. It would help the reader if a column with p values was added.

We have added a column of p-values.

5. Except for a statement on line 299, the authors are largely silent on monetary and 

other incentives to participate in each study. Incentives may be critical to decisions to participate in trials, particularly in LMIC. If data were available for even a subset of included articles, consent by incentives this should be included as sub-analysis. If not, please address as a limitation of the review.

We agree that incentives may be one reason for higher consent rates in LMICs; we addressed this in lines 293-295 of the discussion, which read as follows:

“Furthermore, even with standard of care or placebo control arms, participants may perceive additional incentives such as free health care, increased surveillance and reimbursement of travel costs as benefits to their participation. These incentives may hold greater importance as positive motivators for consent in LMICs.” 

Per your suggestion, we have also added this to the limitations of our study on lines 311-313:

“We cannot comment on other factors beyond trial characteristics that have been associated with informed consent rates such as parental characteristics, physicians approaching participants for consent, or incentives to participation.”

6. There may be a missing reference on line 285.

Thank you for pointing this out. We have double checked that all intended references were included and deleted “[ref]” in the text.

---

## [Editor Report · Decision Letter 1]

24 Feb 2021

Informed consent rates for neonatal randomized controlled trials in low- and lower middle-income versus high-income countries: A systematic review

PONE-D-20-21677R1

Dear Dr. Patterson,

We’re pleased to inform you that your manuscript has been judged scientifically suitable for publication and will be formally accepted for publication once it meets all outstanding technical requirements.

Kind regards,

Ghislaine JMW van Thiel

Academic Editor

PLOS ONE
---

## [Editor Report · Acceptance letter]

26 Feb 2021

PONE-D-20-21677R1 

Informed consent rates for neonatal randomized controlled trials in low- and lower middle-income versus high-income countries: A systematic review 

Dear Dr. Patterson:

I'm pleased to inform you that your manuscript has been deemed suitable for publication in PLOS ONE. Congratulations! Your manuscript is now with our production department. 

Kind regards, 

on behalf of

Dr. Ghislaine JMW van Thiel 

Academic Editor

PLOS ONE